# Endoscopic Ultrasound-Guided Locoregional Treatments for Pancreatic Neuroendocrine Neoplasms

**DOI:** 10.3390/curroncol32020113

**Published:** 2025-02-16

**Authors:** Graziella Masciangelo, Davide Campana, Claudio Ricci, Elisa Andrini, Emilija Rakichevikj, Pietro Fusaroli, Andrea Lisotti

**Affiliations:** 1Gastroenterology Unit, Hospital of Imola, University of Bologna, 40026 Bologna, Italy; masciangelo.lm@gmail.com (G.M.); emilija.rakichevikj@studio.unibo.it (E.R.); pietro.fusaroli@unibo.it (P.F.); 2Department of Medical and Surgical Sciences (DIMEC), Medical Oncology Unit, Alma Mater Studiorum-University of Bologna, IRCCS Azienda Ospedaliero-Universitaria di Bologna, 40138 Bologna, Italy; davide.campana@unibo.it (D.C.); elisa.andrini3@unibo.it (E.A.); 3Department of Internal Medicine and Surgery (DIMEC), Alma Mater Studiorum, University of Bologna, 40126 Bologna, Italy; claudio.ricci6@unibo.it

**Keywords:** neuroendocrine neoplasms, NENs, neuroendocrine tumors, NET, ablation, radiofrequency, EUS-RFA, ethanol, insulinoma

## Abstract

Pancreatic neuroendocrine neoplasms (pNENs) represent approximately 2% of all solid pancreatic tumors. The incidence of pNENs has been increasing in the last decade. The clinical manifestations of pNENs range from hormone secretion syndromes in functioning neoplasms (F-pNENs) to local infiltration or distant metastases in late-stage diagnoses or incidental findings in small non-functioning neoplasms (NF-pNENs). While surgery is the gold-standard treatment for larger and more aggressive tumors, small and low-grade tumors (G1) may be followed-up due to the indolent course of disease. Recently, endoscopic ultrasound (EUS)-guided ablative techniques, such as ethanol injection (EUS-EI) and radiofrequency ablation (EUS-RFA), have emerged as promising options for loco-regional ablations in selected cases. Despite promising safety profile and efficacy, high-quality evidence is needed to support their widespread adoption. This article reviews the current state of EUS-guided locoregional therapies, patient selection criteria, procedural details, and associated risks.

## 1. Introduction

Pancreatic neuroendocrine neoplasms (pNENs) represent 2% of all solid pancreatic tumors and despite being considered rare, their incidence has been progressively increasing over the past years [1]. These neoplasms have highly variable clinical behavior and prognosis, influenced by multiple factors. Key determinants include the degree of differentiation—classified as well-differentiated (NETs) or poorly differentiated (neuroendocrine carcinomas, NECs)—as well as grading parameters such as the mitotic index and Ki67 proliferative index, TNM staging, immunohistochemical expression of somatostatin receptors, and additional histological and molecular markers including DAXX/ATRX genes expression. Based on differentiation and grading, pNENs can be categorized into four main groups: NET G1 (well-differentiated, Ki67 < 3%), NET G2 (well-differentiated, Ki67 3–20%), NET G3 (well-differentiated, Ki67 > 20%), and NEC (poorly differentiated) [2]. The clinical presentation of pNENs varies according to their functional status or symptoms related to locoregional infiltration (e.g., jaundice, pain). Functioning pNENs (F-pNENs) represent 10–30% of all pNENs; F-pNENs, such as insulinomas, glucagonomas, and VIPomas, are characterized by their clinical manifestations due to hormone secretion. Otherwise, non-functioning pNENs (NF-pNENs) usually remain silent until they reach a substantial tumor burden or could be incidentally diagnosed in asymptomatic individuals. Recent improvements and widespread use of cross-sectional imaging modalities and endoscopic ultrasound have led to the increased detection of small, non-functioning pNENs [3]. In particular, it has been estimated that the incidence of PanNETs < 2 cm in size increased by 710% over a 20-year period based on a population analysis performed in the United States [4]. Current guidelines consistently recommend surgical resection for pNENs larger than 2 cm, poorly differentiated neoplasms, and in the case of elevated Ki67 proliferation index (G2–G3), lesions involving the main pancreatic duct or with locoregional infiltration. On the other hand, the management of small (<2 cm), low-grade pNENs (Ki67 ≤ 5%, G1) primarily involves clinical and radiological surveillance considering their biological behavior and the high rates of morbidity and mortality related to pancreatic surgery. A recent study showed that patients with small NF-pNENs who underwent pancreatectomy had similar 5-year cancer-specific survival compared with those managed through surveillance [4,5]. Despite recent advances in surgical techniques, pancreatectomy carries significant risks of morbidity, mortality, and long-term pancreatic exocrine and endocrine dysfunctions. Although mortality associated with pancreatic surgery has decreased over the years in high-volume centers, surgery-related complications remain frequent, affecting up to 50% of patients undergoing pancreatectomy. Among the most frequent surgical-related adverse events, the most common is postoperative pancreatic fistula, with an incidence ranging from 3% to 45% even in tertiary high-volume centers [6]. In this context, in selected cases, locoregional ablative therapies have emerged as an alternative strategy. Among these, endoscopic ultrasound (EUS)-guided techniques, such as ethanol injection (EUS-EI) and radiofrequency ablation (EUS-RFA), have been the most studied methods with promising clinical outcomes. This article aims to present an updated review of the latest evidence on EUS-guided locoregional treatments for pNENs.

## 2. Indications for Locoregional Therapy and Patient Selection

Although several clinical studies were designed to assess the efficacy and safety of EUS-guided ablative techniques, strong evidence supporting their routine use in pNENs is still lacking. For example, the latest European NeuroEndocrine Tumor Society (ENETS) (Berlin, Germany) guidelines do not include locoregional therapies in the therapeutic algorithm [3]. According to ENETS guidelines, a 15–20 mm tumor size appears to be the most reliable cut-off for distinguishing between indolent and aggressive forms. Conservative management through active imaging-based surveillance is considered safe in the short-term for small (<15 mm), asymptomatic pNENs. A watchful strategy should be considered for selected patients, while surgical resection remains the preferred option for young and healthy individuals due to the lack of robust long-term follow-up data. However, in young patients with MEN1 syndrome and non-functioning pNENs smaller than 2 cm, an observational approach is well-established, with long-term studies confirming its safety [7]. Recently, the European Neuroendocrine Tumor Society (ENETS) has published a paper in which they indicate EUS-RFA as a possible treatment for insulinomas ≤ 2 cm in patients unfit for surgery in experienced centers, with a low grade of recommendation [3,8]. Over the past years, two studies have compared locoregional therapies with surgical resection in patients with F-pNENs. In the first study, researchers from the Asan Medical Center in South Korea conducted a retrospective analysis of patients with insulinomas treated between 2011 and 2018, comparing outcomes between those who underwent EUS-EI and those who had pancreatectomy. The authors demonstrated similar 10-year disease-free and overall survival in both groups. However, patients who underwent EUS-EI experienced fewer adverse events and shorter hospitalization [9]. In 2023, Crinò et al. published a multicenter retrospective study comparing EUS-RFA with surgical resection in patients with pancreatic insulinomas [10]. The authors reported similar clinical efficacy between the two approaches but observed a better safety profile in patients treated with EUS-RFA. The authors highlighted that previous EUS-RFA did not preclude patients from undergoing subsequent pancreatectomy in cases of tumor recurrence. Based on these findings, the authors designed a prospective, randomized trial that is currently enrolling patients from multiple centers in order to provide high-quality and robust evidence in this field [11]. While in the case of symptomatic F-pNENs different guidelines give a clear treatment indication, the management of small NF-pNENs is more challenging and should be based on tumor size, grading, lymph node metastasis risk, patient clinical condition, comorbidities, and preferences. Currently, no rigorous prospective study has been conducted to compare surgical resection with active surveillance in this patient subset. As a result, the role of locoregional ablative treatment for NF-pNENs remains unclear; indeed, there may be a risk of overtreatment in indolent tumors with a low risk of progression or undertreatment in aggressive tumors with lymph node involvement [12].

The indications for EUS loco-regional ablations adopted in most studies and ongoing trials are summarized in Table 1.

## 3. Endoscopic Ultrasound-Guided Ethanol Injection Ablation (EUS-EI)

Locoregional ablation of pancreatic neoplasms with ethanol injection at 80–100% concentrations was achieved through coagulative tumor necrosis consequent to cellular and protein dehydration and vascular occlusion [14].

### 3.1. Procedure

EUS-EI can be performed under sedation or general anesthesia, with anesthesiological assistance whether possible, and does not require radiological guidance. The procedure employs a linear-array echoendoscope to identify the optimal window for lesion puncture. A fine needle aspiration (FNA) needle is introduced, avoiding vascular structures, previously identified using Doppler imaging or other interposing structures. The use of needles with several lateral lumes is not recommended due to the risk of ethanol diffusion into the surrounding pancreatic tissue.

After inserting the FNA needle into the center of the lesion, 10 mL of 80–100% ethanol is injected; repeated injections can be performed until the whole lesion appears hyperechoic, depending on the lesion size and ethanol distribution. Before needle withdrawal, a dwell time of 60 s is recommended to minimize ethanol leakage into adjacent healthy tissue and the gastrointestinal wall.

### 3.2. Safety and Adverse Events

Up to 10% of the patients presented EUS-EI-related adverse events; common adverse events are abdominal pain, acute pancreatitis, peritonitis, and venous thrombosis. These complications are attributed to the ethanol-induced inflammatory process into adjacent pancreatic tissue and vascular structures [15].

### 3.3. Clinical Evidence

The first EUS-EI case, reported by Jürgensen et al. in 2007, involved a 13 mm pancreatic tail insulinoma treated with 95% ethanol; the authors reported clinical, biochemical and complete radiological response [16]. Recently, the same group retrospectively analyzed 33 patients suffering from pancreatic insulinoma; over an 11-year study period, they compared EUS-EI (9 patients) to surgery (24 patients). They observed similar clinical efficacy but a significantly lower incidence of adverse events in the EUS-EI group (11% vs. 54%) [13].

The main evidence in the field of EUS-EI is summarized in Table 2 (lesion size is expressed in terms of mm, the median ± standard deviation, or the median [range]). No prospective study comparing EUS-EI with surgery or follow-up is available.

## 4. Endoscopic Ultrasound-Guided Radiofrequency Ablation (EUS-RFA)

Treatment of pancreatic neoplasms with radiofrequency ablation involves the delivery of a high-frequency alternating current within the tumor through an electrode needle. This electrode introduces focused energy, precipitating localized high-current density that results in high local temperatures that induce tissue coagulative necrosis (direct effect). Additionally, a T-cell mediated response has also been suggested as an immune-modulatory mechanism of action (indirect effect) [25].

EUS-RFA should be preferentially performed in tertiary centers with expertise in advanced endoscopic procedures and a dedicated multidisciplinary team on pNENs. Ideally, interventional radiology and an endoscopic retrograde cholangiopancreatograpy (ERCP) service should be available on-site or on-call to manage any vascular or pancreatic adverse event such as bleeding or pancreatic duct injury. In addition, access to a dedicated pancreatic surgical team is suggested for prompt intervention if conversion to an operative procedure becomes necessary. Given the potential risks, these procedures are generally performed on an inpatient basis, allowing for at least 24-h post-procedural monitoring to detect complications like pancreatitis, ductal injuries, or bleeding. Post-procedural pain control is suggested with NSAIDs or analgesics. After the procedure, a structured follow-up is recommended including clinical evaluation and imaging studies (e.g., contrast-enhanced CT or repeat EUS) to assess the treatment efficacy and monitor for recurrence or delayed complications. Unfortunately, we are facing a dramatic lack in standardization of pre-procedural, procedural, and post-procedural management, suggesting the need for action to overcome this limit such as a consensus conference or an international survey of experts.

### 4.1. Procedure and Device

Two types of RFA systems are available in Europe: the through-the-needle devices, such as the Habib™ EUS-RFA catheter, inserted via FNA needles (19–22 gauge), or needle devices, such as the STARmed EUSRA needle, featuring a 19-gauge electrode with an internal saline cooling system to prevent tissue carbonization [8].

The STARmed system connects the 19-gauge EUSRA needle to a dedicated impedance-controlled generator (VIVA Combo RF generator; Taewoong Medical) with an integrated cooling system [8].

The needle is advanced into the distal portion of the lesion under continuous EUS control. The energy delivery (10–50 Watts) lasts for a few seconds and is automatically interrupted when the temperature or impedance thresholds are reached. Multiple ablations within the lesion may be performed to achieve complete treatment. Precautions are taken to prevent thermal injury to adjacent healthy tissue or the gastrointestinal wall by positioning the needle at the lesion’s distal margin and delaying device withdrawal after energy application. The use of contrast-enhanced harmonic with the injection of an ultrasound contrast agent is recommended to assess the residual vital tissue after ablation [26,27]. Schematic representations of an EUS-RFA procedure for a 7 mm F-pNEN of the pancreatic neck is reported in Figure 1.

The Habib EUS-RFA catheter is a monopolar electrode (1 Fr × 220 cm) designed for use within a 19- or 22-gauge FNA needle. After introducing the needle into the target lesion, the stylet is removed and replaced with the catheter. The catheter is advanced through the needle until resistance is encountered, indicating contact with the tissue.

To achieve a correct device placement, the catheter is held stationary while the FNA needle is gradually retracted by approximately 3 cm, exposing the tip of the catheter. In a single procedure, up to 10 ablations can be carried out, with each ablation lasting approximately 90–120 s and delivering 10–15 kW of energy. In the absence of an integrated cooling system within the probe, some authors have recommended waiting at least 60 s before successive applications to prevent overheating. Currently, the Habib RFA probe is not available on the market [8].

Prophylactic administration of rectal NSAIDs and aggressive hydration is suggested to reduce the risk of post-procedure acute pancreatitis. On the other hand, no consensus on the indication for antibiotic prophylaxis has been reached yet [28,29].

A large French study identified a distance ≤ 1 mm between the lesion and the main pancreatic duct as an independent risk factor for adverse events post-EUS-RFA. Close proximity between the pNEN and pancreatic duct increased the risk of acute pancreatitis and pancreatic duct stenosis [19]. Therefore, a distance greater than one millimeter is recommended for safe application. In the case of pNENs that are in close proximity to the main pancreatic duct, some operators suggest prophylactic pancreatic stenting through endoscopic retrograde cholangiopancreatography (ERCP); the authors suggest performing stenting several days prior to EUS-RFA to prevent ductal injuries. However, there is no evidence to support this approach and consensus in this field is lacking [28,29].

### 4.2. Safety and Adverse Events

EUS-RFA is generally considered as a safe and well-tolerated procedure. Barthet et al. reported two major adverse events, namely severe acute pancreatitis and duodenal perforation in the first two patients who underwent pancreatic EUS-RFA. The authors adopted a pre-procedure protocol involving rectal NSAIDs and aggressive intravenous hydration that led to a dramatic reduction in the incidence of severe adverse events [24].

To date, the most frequently reported adverse events are mild, and among these, the most common are abdominal pain, fever, and mild acute pancreatitis. Severe complications, such as ductal stenosis or injuries leading to peripancreatic fluid collections, are rare. However, standardized diagnostic criteria for post-RFA pancreatitis are lacking and an overestimation of the incidence cannot be excluded when applying the Atlanta criteria.

### 4.3. Clinical Evidence

The main evidence on the outcomes of EUS-RFA for pNEN treatment is reported in Table 1. A recent meta-analysis including 11 studies with 292 patients and a minimum follow-up of 12 months reported a complete radiological response in 87.1% of cases and a partial response in 11.4%. Among patients with F-pNENs, the clinical response was 94.9%. The overall adverse event rate was 20%, with a severe event rate of 0.9% and no procedure-related mortality [28,29].

Long-term data on the outcomes of EUS-RFA for pNENs showed stable responses in 12 patients with a median follow-up of 43 months [30]; a complete radiological response with tumor disappearance was achieved in 85.7%, while one patient developed liver metastasis after the failure of EUS-RFA.

Two prospective cohort studies are currently in progress, registered on ClinicalTrials.gov, for the further safety and efficacy profile assessment of EUS-RFA in NF-pNENs: RAPNEN (NCT03834701) and RFANET (NCT04520932).

## 5. Endoscopic Ultrasound-Guided Microwave Ablation (EUS-MWA)

Microwave ablation (MWA) generates intralesional heat through dipole molecular oscillation, resulting in controlled temperature elevation up to 90 °C. This technique delivers deep, uniform, and consistent energy, inducing coagulative necrosis similar to RFA but with minimal cooling time requirements. Unlike RFA, MWA does not rely on tissue impedance, is not constrained by water vaporization or carbonization, and does not require an electrical circuit [31].

### 5.1. Procedure and Device

Currently, this type of device is not available in Europe. Further technical and clinical development is required for broader implementation.

### 5.2. Clinical Evidence

In 2022, a case report detailed the use of EUS-MWA for treating an unresectable pNEN, achieving complete lesion ablation with a sustained response over an 8-month follow-up. During the procedure, a linear-array echoendoscope was used coupled with a dedicated microwave energy platform (CROMA, Creo Medical, Chepstow, Wales, UK) and a 19.5-gauge needle antenna (MicroBlate Fine, Creo Medical, Wales, UK) [31].

## 6. Percutaneous Image-Guided Ablation Techniques for pNENs

In addition to EUS-guided loco-regional ablation techniques, percutaneous approaches, such as computed tomography (CT) or ultrasound (US)-guided RFA and percutaneous ethanol ablation, have been reported in selected patients with pNENs. Although the experience with these techniques is even more limited than with EUS-guided methods, they may be considered when lesions are favorably located (i.e., in the pancreatic body or tail and clearly visible on cross-sectional imaging) or when endoscopic access is challenging. Percutaneous RFA offers the advantage of widely available CT or US guidance and may be particularly useful for lesions distant from the gastrointestinal lumen [32,33]. However, concerns about pancreatic injury, needle tract seeding, and periprocedural pancreatitis remain. Similarly, percutaneous ethanol ablation has been applied in isolated cases but carries the risk of ethanol leakage and consequent tissue injury. In clinical practice, a decision algorithm that integrates tumor size, location, proximity to the main pancreatic duct, and patient-specific factors (i.e., anesthesiological risk) can help guide the choice between EUS-guided and percutaneous ablation. Until robust comparative studies are available, percutaneous ablation should be reserved for highly selected cases, and its use should be weighed against the current standards of minimally invasive surgery and watchful waiting, as outlined in the ENETS and NANETS guidelines [3,34].

## 7. Conclusions

In conclusion, preliminary findings suggest that EUS-guided loco-regional ablation techniques could be considered as safe and effective therapeutic options in the treatment of patients suffering from pNENs. Current ENETS and North America NeuroEndocrine Tumor Society (NANETS) guidelines strongly recommend surgical resection for neoplasms larger than 2 cm, those with aggressive features (i.e., G2–G3, Ki-67 > 5% or local invasion), or in the case of F-pNENs causing symptoms due to hormonal secretion [3,34]. In this field, minimally invasive (laparoscopic or robotic) pancreatic surgery has improved the safety outcomes; however, pancreatectomy is still burdened by significant morbidity including pancreatic fistula, diabetes, and exocrine insufficiency [6]. On the other hand, watchful follow-up is an accepted strategy for small (≤2 cm), asymptomatic, low-grade NF-pNENs, as demonstrated by several studies suggesting a low risk of progression or cancer-related mortality over time [3,4]. When considering treatment options for small pNENs, EUS loco-regional ablation could be taken into account as an alternative therapeutic option in comparison with surgical resection and active surveillance. EUS-guided loco-regional ablation presents a potential middle ground—offering a non-surgical approach that avoids the risks of pancreatectomy while addressing tumors that may otherwise progress during surveillance. However, its role remains undefined due to a lack of long-term oncological data. Until robust prospective trials confirm the safety, EUS ablation should be reserved for selected patients such as those unfit for surgery or with F-pNENs requiring symptom control. Although many studies support the safety and technical feasibility of these techniques, robust trials are needed to provide evidence that can support the clinical indication and standardized application protocols. In particular, EUS-RFA seems to represent a valid therapeutic option for pNENs, especially in patients who present secretory symptoms resulting from F-pNENs, and in high-risk surgical patients.

## Figures and Tables

**Figure 1 curroncol-32-00113-f001:**
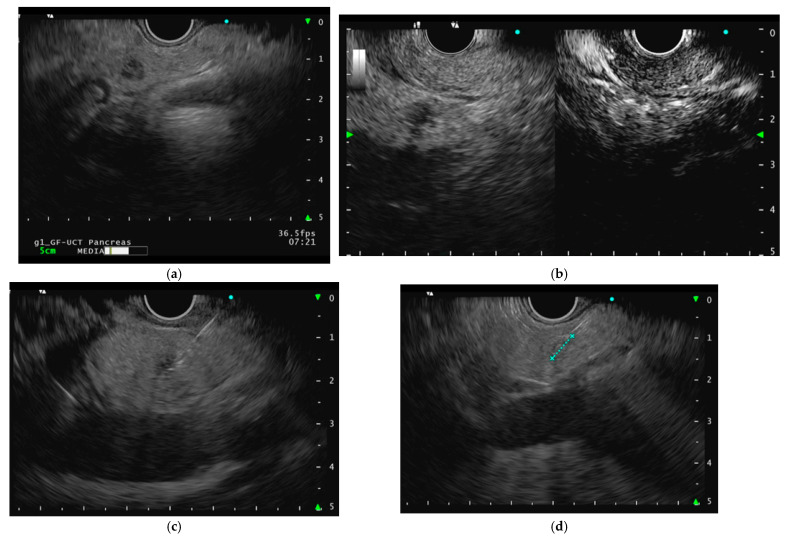
(**a**) Endoscopic ultrasound image of a small hypoechoic insulinoma (functioning pancreatic neuroendocrine neoplasm) of the pancreatic neck; (**b**) contrast-enhanced harmonic endoscopic ultrasound (CH-EUS) after the injection of an ultrasound contrast agent showing a homogeneously hyper-enhanced neoplasm during the arterial phase, with a peripheral rim; (**c**) the 7 mm endoscopic ultrasound radiofrequency ablation probe (EUSRA, Taewoong Medical) insertion; (**d**) accurate tip control under endoscopic ultrasound evaluation; (**e**) EUS-RFA application at 40 Watt power setting, confirmed by the appearance of hyperechoic bubbles suggesting coagulative necrosis; (**f**) contrast-enhanced harmonic endoscopic ultrasound (CH-EUS) after the injection of an ultrasound contrast agent showing complete ablation with no residual vital tissue.

**Table 1 curroncol-32-00113-t001:** Treatment indications for pNENs based on size, grade, and functional status.

Tumor Size	Tumor Grade	Functional Status	Recommended Treatment Approach	Comments/References
<15 mm	G1	Non-functioning	**Active surveillance** is generally preferred. EUS-guided ablation may be considered in selected high-risk patients (e.g., elderly or those with significant comorbidities).	ENETS/NANETS guidelines favor surveillance for small, indolent lesions; see also the inclusion criteria of RAPNEN/RFANET (which focus on non-functioning pNENs ≤ 2 cm) for further evaluation in selected cases.
15–20 mm	G1	Functioning (insulinoma)	**EUS-guided ablation** is a viable option to control secretory symptoms, particularly in patients who are poor surgical candidates or who prefer a less invasive approach.	Studies by Jürgensen et al. and Crinò et al. [10,13] have shown promising results in F-pNENs; while surgery remains the standard, ablation may offer lower morbidity and shorter hospital stays in select cases.
15–20 mm	G1	Non-functioning	**Surgical resection** is typically recommended; however, EUS-guided ablation may be considered for patients unfit for surgery. Ongoing trials (RAPNEN, RFANET) are assessing efficacy in this group.	While active surveillance is an option in some cases, patient factors (e.g., age, comorbidities) and lesion location may tip the balance toward intervention. Current studies are exploring ablation outcomes in NF-pNENs.
>20 mm	Any (or if Ki67 is elevated)	Either functioning or non-functioning	**Surgical Resection** is the treatment of choice due to the higher risk of aggressive behavior, local invasion, or metastasis.	Robust evidence and current guidelines (ENETS, NANETS) support surgery for larger or higher-grade lesions; locoregional ablation is generally not indicated in these cases.

**Table 2 curroncol-32-00113-t002:** Main evidence in the field of endoscopic ultrasound locoregional ablation for pancreatic neuroendocrine neoplasms.

Authors, Reference	Study Design	Population	pNEN (F/NF)	Lesion Size (mm)	Technical Success	Clinical Success *	Adverse Events (Number)	Adverse Events
**Ethanol** **injection (EUS-EI)**								
Jurgensen C et al. Ultraschall Med 2024 [13]	Retrospective, comparative vs. surgery	9 vs. 24	9/0 vs. 24/0	13.9 ± 7.3	100%	89%	1	Mild pancreatitis
Choi JH et al. Dig Endosc 2018 [17]	Prospective	33	1/39	11 [7–20]	100%	60%	2	Mild and moderate pancreatitis
Park DH et al. Clin Endosc 2015 [18]	Retrospective	11	4/10	12.3 ± 3.7	100%	75%	11	Mild pancreatitis
**Radiofrequency ablation (EUS-RFA)**								
Crinò SF et al. Clin Gastroenterol Hepatol 2023 [10]	Retrospective, comparative vs. surgery	89 vs. 89	89/0 vs. 89/0	13.4 ± 3.9	100%	95.5%	16	Acute pancreatitis
Napoleon N et al. Gastrointest Endosc. 2023 [19]	Retrospective	64	16/48	15.0 [5–30]	100%	87.5%	21	Epigastric pain, Acute pancreatitis, Main pancreatic duct leak
Borelli de Andreis F et al. Pancreatology 2023 [12]	Retrospective	10	10/0	11.9 ± 3.3	100%	100%	2	Abdominal pain
Rizzatti G et al. Endoscopy (suppl) 2023—Abstract ESGE Days [20]	Prospective	56	24/32	NR	100%	100%	8	Acute mild pancreatitis, pancreatic hematoma, bleeding, pancreatic moderate duct injury
Marx M et al. Gastrointest Endosc. 2022 [21]	Retrospective	27	0/27	14.0 ± 4.6	100%	---	4	Not reported
De Nucci G et al. Endosc Int Open 2020 [22]	Prospective	10	0/11	14.5 [9–24]	100%	---	2	Abdominal pain
Oleinikov K et al. J Clin Endocrinol Metab 2019 [23]	Retrospective	18	7/11	14.3 + 7.3	100%	Not reported	0	Not reported
Barthet M et al. Endoscopy 2019 [24]	Prospective	12	0/14	13.1 [10–20]	100%	---	3	Pancreatitis, small bowel perforation

* Clinical success was defined as resolution of hypoglycemia in patients who underwent endoscopic ultrasound radiofrequency ablation for insulinomas.

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
