# Peer review of "Endoscopic Ultrasound-Guided Locoregional Treatments for Pancreatic Neuroendocrine Neoplasms"

_curroncol, 2025, doi:10.3390/curroncol32020113_

Round 1
Reviewer 1 Report
Comments and Suggestions for Authors
I have read the review with interest. Congratulations to the authors.
A table could be added summarizing the treatment indications based on the size of the tumor, the grade, and the patient's functional status, providing a clear guide for clinical decision-making.
Minor revision:
Remove the acronym from line 74 and write the full name
Author Response
We sincerely thank the academic editor for the precious assessment and insightful suggestions made on our manuscript.
Enclosed please find the revised version of our manuscript, according to these considerations.
We made a point-by-point list of all replies and changes we made according to the academic editor's comment.
Academic editor: The data on this very special therapeutic approach is still quite scarce. It is worth to be summarized as you did but conclusions have to be drawn with great caution.
Re: We agree with Editor's view that, despite preliminary interesting data, robust evidence on the role of EUS ablation for pNENs is still lacking. We tone down the conclusion and abstract accordingly.
Academic editor: Espacially the indication part should be more elaborated. Which tumors are suitable regarding technical success, risk/safety, and oncological rationale. Size and location is also important in this section. When it comes to alternatives: What about them? Is it necessary at all to ablate these tumors? Refer to the current guidelines (ENETS, NANETS) and elaborate a potential role for EUS-ablation (vs. minimally invasive/robotic surgery and vs. watchful waiting).
Re: We thank the academic editor for the opportunity of clarifying one of the most debated issue regarding local ablation, compared to other approaches, namely surgery, minimally-invasive surgery, clinical follow-up. Since this issue is still debated, we tried to summarize Guidelines indications together with the inclusion criteria and indication for EUS ablation adopted in published manuscripts.
Academic editor: Are there percutaneous approaches for local ablation worth mentioning? Propose if you like a decision algorithm or a table with the relevant factors.
Re: According to this insightful suggestion, we searched for all other ablative approaches for pNENs: data in this field are even scanter, but we summarized all available approaches in a dedicated paragraph. Since no comparative data is available, we cannot propose a therapeutic algorithm; we included factors that should be taken into account in this paragrah.
Academic editor: Are there any minimal requirements to do the procedure? For example, to have an interventional radiologist at hand if it comes to bleeding complications). What is the recommendation for follow-up. In-patient procedure? Pain management during and after the procedure?
Re: In this field there is a dramatic lack of standardization, and we tried to underline it in the last paragraph; we also suggested potential approaches to overcome this limit, like creating a methodology working group, an international survey of experts, and a Delphi consensus.
Academic editor: In the conclusion, you were quite careful, which is OK. But also mention the alternative management backed up by current guidelines.
Re: Once again, we sincerely thank the academic editor for the positive assessment of our manuscript. In this revised version of the submitted manuscript, we tried to expand the point of view, including also other alternative approaches for management, treatment, and ablation of pNENs.
Reviewer #1: I have read the review with interest. Congratulations to the authors.
A table could be added summarizing the treatment indications based on the size of the tumor, the grade, and the patient's functional status, providing a clear guide for clinical decision-making.
Re: We thank Reviewer #1 for the positive assessment. We made the corrections suggested.
Reviewer #1: Minor revision: Remove the acronym from line 74 and write the full name
Re: Done
Reviewer #2: This review study was aimed to investigate current status of endoscopic ultrasound-guided local ablation therapies for pancreatic NEN, especially in pancreatic NEN to be not suitable for absolute surgical indication. Nowadays, small and low-grade p-NEN has been considered to be treated by observation follow-up without any interventional therapeutic approach.
Recent several trial such as endoscopic ultrasound-guided local ablation therapies have been reported. However, there are no clearly identified evidence with high levels about it. Therefore, authors investigate the review study on it. The manuscript was well-investigated and wee-written on it.
Re: We thank Reviewer #2 for the positive assessment. We made the corrections suggested.
Reviewer #2: Minor revision: 1, In Table 1, lesion size was shown in each literature. The size shown in this Table might be median or mean diameter. It should be clarified about it. And also range of lesion size should be shown if possible.
Re: Table has been changed and the requested info has been included. In the text, we also specified “(lesion size is expressed in terms of mm, as median ± standard deviation or as median [range])”.
Reviewer #3: The manuscript is well-done. The topic is relevant and interesting. However the conclusion is too brief, it should be more detailed, regarding the summary of the indications of the EUS-guided loco-regional modalities.
Re: We thank Reviewer #3 for the positive assessment. We made the corrections suggested; in detail, we expanded the point of view, including also one of the most debated issue regarding local ablation. We compared the indication and results to other approaches, namely surgery, minimally-invasive surgery, clinical follow-up, an even percutaneous approaches. Since this issue is still debated, we tried to summarize Guidelines indications together with the inclusion criteria and indication for EUS ablation adopted in published manuscripts.
Yours sincerely,
Andrea Lisotti
Reviewer 2 Report
Comments and Suggestions for Authors
This review study was aimed to investigate current status of endoscopic ultrasound-guided local ablation therapies for pancreatic NEN, especially in pancreatic NEN to be not suitable for absolute surgical indication. Nowadays, small and low-grade p-NEN has been considered to be treated by observation follow-up without any interventional therapeutic approach.
Recent several trial such as endoscopic ultrasound-guided local ablation therapies have been reported. However, there are no clearly identified evidence with high levels about it. Therefore, authors investigate the review study on it. The manuscript was well-investigated and wee-written on it.
There was just a minor points below to be corrected.
Minor point;
1, In Table 1, lesion size was shown in each literature. The size shown in this Table might be median or mean diameter. It should be clarified about it. And also range of lesion size should be shown if possible.
Author Response

(The authors gave the same response as above.)

Reviewer 3 Report
Comments and Suggestions for Authors
The manuscript is well-done. The topic is relevant and interesting. However the conclusion is too brief, it should be more detailed, regarding the summary of the indications of the EUS-guided loco-regional modalities.
Author Response

(The authors gave the same response as above.)
